# Hands Ahead in Mind and Motion: Active Inference in Peripersonal Hand Space

**Johannes Lohmann, Anna Belardinelli**  **and Martin V. Butz**

Cognitive Modeling, Department of Computer Science, Faculty of Science, University of Tübingen, 72076 Tübingen, Germany; belardinelli@uni-tuebingen.de (A.B.); martin.butz@uni-tuebingen.de (M.V.B.)
* Correspondence: johannes.lohmann@uni-tuebingen.de

**Abstract:** According to theories of anticipatory behavior control, actions are initiated by predicting their sensory outcomes. From the perspective of event-predictive cognition and active inference, predictive processes activate currently desired events and event boundaries, as well as the expected sensorimotor mappings necessary to realize them, dependent on the involved predicted uncertainties before actual motor control unfolds. Accordingly, we asked whether peripersonal hand space is remapped in an uncertainty anticipating manner while grasping and placing bottles in a virtual reality (VR) setup. To investigate, we combined the crossmodal congruency paradigm with virtual object interactions in two experiments. As expected, an anticipatory crossmodal congruency effect (aCCE) at the future finger position on the bottle was detected. Moreover, a manipulation of the visuo-motor mapping of the participants' virtual hand while approaching the bottle selectively reduced the aCCE at movement onset. Our results support theories of event-predictive, anticipatory behavior control and active inference, showing that expected uncertainties in movement control indeed influence anticipatory stimulus processing.

**Keywords:** event-predictive cognition; crossmodal congruency; peripersonal space; virtual reality

## 1. Introduction

Over the recent decades, the nervous system has been progressively viewed as a predictive inference machine, rather than a mere information processor [1–3]—a perspective that dates back to Hermann von Helmholtz, who considered perception to be an inference process. The central claim is that the central nervous system anticipates future states in order to support goal-directed, self-motivated behavior [4–8]. Hence, the initiation of goal-directed interactions is preceded by activating future state estimations. This prediction dates back to the ideomotor principle [9], but also closely matches more recent neurocomputational formalizations of active inference processes based on free energy formalizations [10–12], which yield anticipatory (i) future state activations and (ii) focused sensorimotor processing dependent on currently expected, task-relevant uncertainties. Active inference formalizes two essential organismic needs. First, the reduction of expected uncertainties, leading to information-seeking behavior, and second, the reduction of expected discrepancies between predicted and desired futures. When following this principle, as a result organisms act in a goal-directed, uncertainty-minimizing manner. However, to be able to apply active inference effectively, suitable predictive encodings are required.

The representational format of these predictive encodings is still being debated. It has been proposed that actions and perceptions are integrated into common event codes [6]. On more abstract levels, video streams have been shown to be systematically segmented into events and event boundaries [13,14]. Thus, it appears that event-predictive structures (EPSs) are formed on numerous levels of abstraction [14,15], encoding sensorimotor changes but also the expected final outcome of particular sensorimotor

interactions and contextual pre-conditions. Given such EPSs, active inference processes can be assumed to activate expected final outcomes and necessary sensorimotor dynamics in a goal-directed manner. Furthermore, they will focus sensorimotor processing on those anticipated events and event boundaries that are expected to be most important to ensure behavioral success.

Behavioral evidence for such an event-oriented active inference process comes from eye-tracking studies. There are many examples that show how task demands shape fixation patterns [16,17]. Belardinelli, Stepper, and Butz [18] showed a preference of eye fixations for the future index finger position in a grasping task in anticipation of the targeted, final object manipulation (e.g., drinking from or handing over a bottle). Apparently, visual processing was tuned to critical spatial locations to ensure a successful, intended object manipulation, which dovetails with the proposed event-predictive, active inference process.

Besides visual processing, the representation of the space surrounding the body—that is, peripersonal space (PPS) and particularly peripersonal hand space (PPHS)—appear crucial for successful object interactions and adaptive behavior in general [19,20]. PPS has been shown to be quintessentially multisensory [21–23]. This integration seems to be realized by neurons in the ventral intraparietal area [24]. In humans, PPHS is typically investigated by means of a crossmodal congruency paradigm [25]. Participants have to report the location of a tactile stimulation as fast as possible while task-irrelevant visual stimuli are presented. When applied close to the stimulated body parts, the visual stimuli have been shown to selectively interact with the tactile perception. For instance, participants were slower to respond to a stimulation of the index finger, or the thumb, if an LED was flashed at the non-stimulated finger (incongruent condition), compared to trials where the location of visual and tactile stimulation approximately coincided (congruent condition). This visuo-tactile integration seems specific to PPHS and the effect decreases with increasing distance between visual and tactile stimuli [25].

The multisensory nature of PPS has been considered to support self-defensive behavior [26], as it could facilitate motor responses to potentially threatening stimuli [19]. However, some researchers stressed that the same mechanism can be used to control goal-directed actions [19,27,28]. In line with this view, different studies showed crossmodal congruency effects at the tip of tools when these were actively used by participants [29,30]. These results imply that the boundaries of PPS were modulated by the action possibilities, remapping PPS to an action-relevant, spatially distant location. While it remains open to which degree shifts in spatial attention alone can account for these findings [30,31], these results show that multisensory processing in spatial body representations is modulated by action possibilities and intentions, mapping the space that can be interacted with onto according behavior [32].

This notion is in line with results from recent studies that have shown anticipatory remapping of PPHS during goal-directed object grasps [33,34] and pantomimic object manipulations [35]—light stimuli at the target object, close to where the fingers will get in contact with it, selectively interacted with the perception of vibrotactile stimuli on the fingers even before the hand started to move toward the target. This anticipatory crossmodal congruency effect (aCCE) indicates an anticipatory remapping of PPHS in the service of action control. In earlier studies regarding this aCCE, participants had to perform instructed interactions with the target object, which remained visible throughout the experiment [33,34]. More recently, Belardinelli et al. [35] showed that the aCCE also occurs during object manipulation tasks and when the object becomes visible only shortly before the vibrotactile stimuli on the fingers and the interacting visual stimuli at the target object are applied. Participants had to perform a pantomimic grasp of a bottle. The orientation of the bottle called for an overhand or an underhand grasp, dependent on if the bottle was oriented upright or upside down. This is due to the so-called end-state comfort effect [36,37]. The results showed that even when the bottle orientation varied unpredictably from trial to trial, a stable aCCE was present and systematically depended on the orientation of the bottle (upright or upside down) and the consequent type of grasp (overhand versus underhand). That is, the aCCE was significant with respect to where the index finger and the thumb would be placed on the bottle (right or left side). These findings corroborate evidence that the aCCE

indeed reflects an adaptive remapping of PPHS in the service of behavior control, emphasizing the functional role of PPHS during object interactions [26,38].

Considering the active inference perspective described above, this functional role can be described by a predictive control mechanism that focusses sensorimotor processing on the next task-relevant events and event boundaries. The aCCE allows one to directly probe this mechanism on a sensorimotor level: The remapping of PPHS for the purpose of initiating and controlling an upcoming grasp yields multisensory interactions at the future hand location. This remapping unfolds over time and yields a stronger aCCE at later stimulus onset asynchronies (SOAs). As mentioned above, the unfolding active inference process can be described in terms of a generative process that estimates the likelihood of sensory data [39]. This probabilistic perspective fits with the Bayesian approach put forward by Noel et al. [40]. Noel et al. conceptualized PPS as a stochastic bubble, wherein the probability of contact or impact with the body is computed, specifically accounting for the coupling probability of diverse sensory signals. Since PPS representations are anchored to related body parts, such computations must extend into the future and predict the probability of a sought interaction. In a virtual reality study, the authors showed that this sensory integration can be formalized as a Bayesian inference problem, with two parameters defining the shape and strength of the a-priori coupling of visual and proprioceptive signals.

Here we reasoned that if the aCCE indeed results from an unfolding active inference process, it should depend on both the intended object manipulation and the expected reliability of the mapping between anticipated sensory consequences and motor control commands. Accordingly, we probed this reasoning by manipulating sensorimotor mappings in an object interaction task. Seeing that the manipulation of sensorimotor mappings is difficult to realize in real world setups, we conducted the respective experiments within virtual reality (VR), manipulating the contingency between the actual and seen hand position.

Participants performed a grasp-and-place task in VR, interacting with a virtual bottle. At different times before and during the interaction, participants received a tactile stimulation at the thumb or index finger. Concurrently, a visual stimulus appeared on the left or right side of the bottle, either matching the future location of the stimulated finger, or not. Participants had to respond as fast as possible by verbally naming the finger that was stimulated. We expected faster responses when the visual stimulus matched the future position of the stimulated finger. The first experiment aimed at replicating the previously observed aCCE [35] in VR. In the second experiment, we dissociated seen and felt hand position by introducing a lateral offset of the virtual hand during movement execution, effectively increasing the uncertainty during the reaching event. As a result, following theories of event-predictive, active inference, we expected that this manipulation would affect the magnitude of the aCCE, especially at movement onset, when the discrepancy between vision and proprioception is about to, and anticipated to, become apparent.

## 2. Materials and Methods

### 2.1. Participants

Twenty-three students from the University of Tübingen participated in the first experiment (twelve females). Their age ranged from 19 to 30 years ($M = 21.9$, $SD = 2.9$). All participants were right-handed and had normal or corrected-to-normal vision. Participants provided informed consent and received either course credit or a monetary compensation for their participation. For the second experiment, another twenty-two participants were recruited (twelve females), none of whom participated in the first experiment. Their age ranged from 18 to 27 years ($M = 22.5$, $SD = 2.6$). Again, all participants were right-handed and had normal or corrected-to-normal vision. They provided informed consent and were compensated with course credit or money for their participation. Some participants had difficulties with the virtual grasping procedure and could not complete the experiment; this was true for seven participants (three males) from the first, and for one participant (one male) from the second

experiment. The reduced drop-out-rate was due to a prolonged training phase applied in the second experiment. The respective data were not considered in the analysis.

Sample sizes were determined based on a power analysis using our earlier data regarding aCCE (Experiment 1 in Belardinelli et al. [35]). For the sought three-way interaction, we previously observed an effect sizes of $\eta_p^2 = 0.64$. Given a power of 0.9 and an alpha level of 0.05, a lower bound for the sample size of 20 was determined. The power analysis was performed by means of the Monte Carlo method. R scripts and results for the power analysis are provided in the Supplementary Material (S1).

### 2.2. Apparatus

Participants were equipped with an Oculus Rift© DK2 stereoscopic head-mounted display (HMD; Oculus VR LLC, Menlo Park, CA, USA). The refresh rate was 75 Hz. Motion tracking of hand movements was realized with a Leap Motion© near-infrared sensor (Leap Motion Inc., San Francisco, CA, USA, SDK version 2.3.1). The Leap Motion© sensor provides positional information regarding the palm, wrist, and phalanges with a target refresh rate of 120 Hz. This data can be used to render a hand model in VR. The Leap Motion© sensor provides a fast, calibration-free means to collect hand kinematics—at the cost of reduced spatial and temporal resolution when compared to classic optical tracking systems [41,42]. To allow participants to respond to the tactile stimulation without manual response, participants were equipped with a headset. Speech recognition was implemented by means of the Microsoft Speech API 5.4, which provides a time-stamp at the beginning of an utterance. The whole experiment was implemented with the Unity® engine 5.5.0 (Unity Technologies, San Francisco, CA, USA) using the C# interface provided by the API. During the experiment, the scene was rendered in parallel on the Oculus Rift and on a computer screen, such that the experimenter could observe and assist the participants. Tactile stimulation was realized by means of two small (10 mm × 3.4 mm) shaftless vibration motors (Pololu Corporation, Las Vegas, NV, USA; vibration amplitude of 0.75 g at 3 V, 14500 rotations per minute at 3 V) attached to the tip of the thumb and the index finger of the participants. The motors were controlled via an Arduino Uno microcontroller (Arduino S.R.L., Scarmagno, Italy) running custom C software. The microcontroller was connected to the computer via an USB port, which could be accessed by the Unity® program. The wiring diagram, as well as additional information regarding the components, can be found at the first author's webpage (http://www.uni-tuebingen.de/de/26084).

### 2.3. Virtual Reality Setup

The VR setup called for participants to be placed in an office-like room. Centered about 50 cm in front of them, a pedestal was placed, where the target object appeared during the trials. The target was always a 3D model of a plastic bottle, either oriented upright or upside down. The bottle was 15 cm in height, subtending a visual angle of 17.1° at the initial location. A second pedestal, 15 cm to the right of the first one, served as the target location (see Figure 1). The positions of the pedestals were marked with actual cardboard boxes, providing haptic feedback regarding the bounds of the task space (participants were seated in a way that they had to stretch their arm to reach the pedestals). Instructions and feedback were presented in different text-fields, aligned at eye-height. At the beginning of a trial, a fixation cross appeared at the initial location of the target bottle (see Figure 2, bottom). The fixation cross was 10 cm wide and 10 cm high, subtending a visual angle of 11.4°. The visual distractor was realized by means of a red, spherical flash with a diameter of 8 cm (equal to a visual angle of 8°) appearing at the left or right side of the bottle.

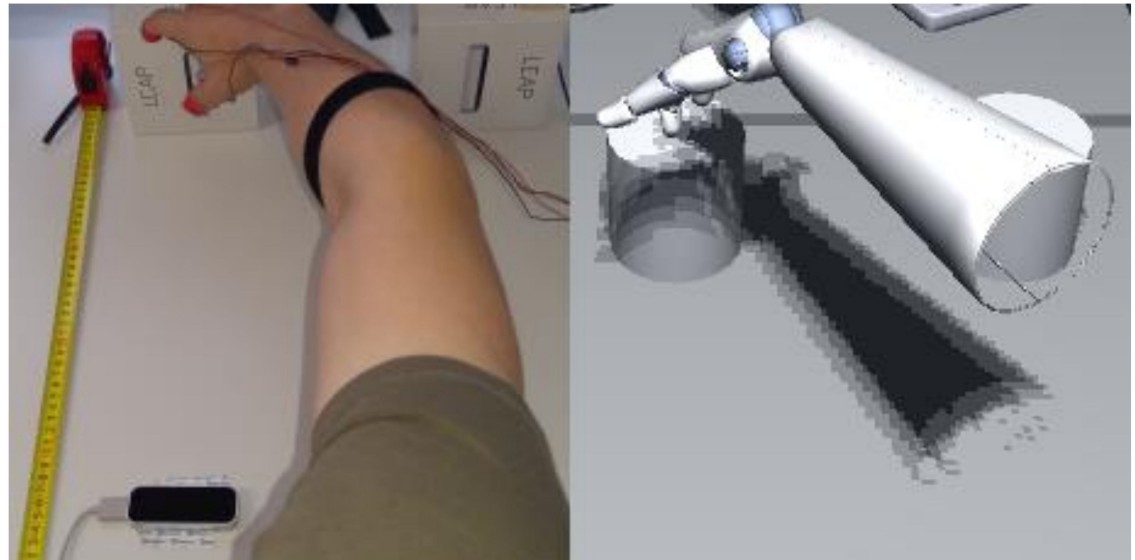

**Figure 1.** Real world setup and according VR impression. Cardboard boxes in the upper part of the left image served as haptic cues for the virtual pedestals representing the target locations for the bottle interaction. The central pedestal was about 50 cm away from participants, the other pedestal was 15 cm to the right and served as target location for the bottle placement. The Leap Motion© sensor was placed in front of the participants, approximately 40 cm before the central pedestal. Vibration motors were attached to the thumb and index finger tips of the participants using Velcro stripes. The motors were controlled with an Arduino Uno microcontroller (not shown in the image).

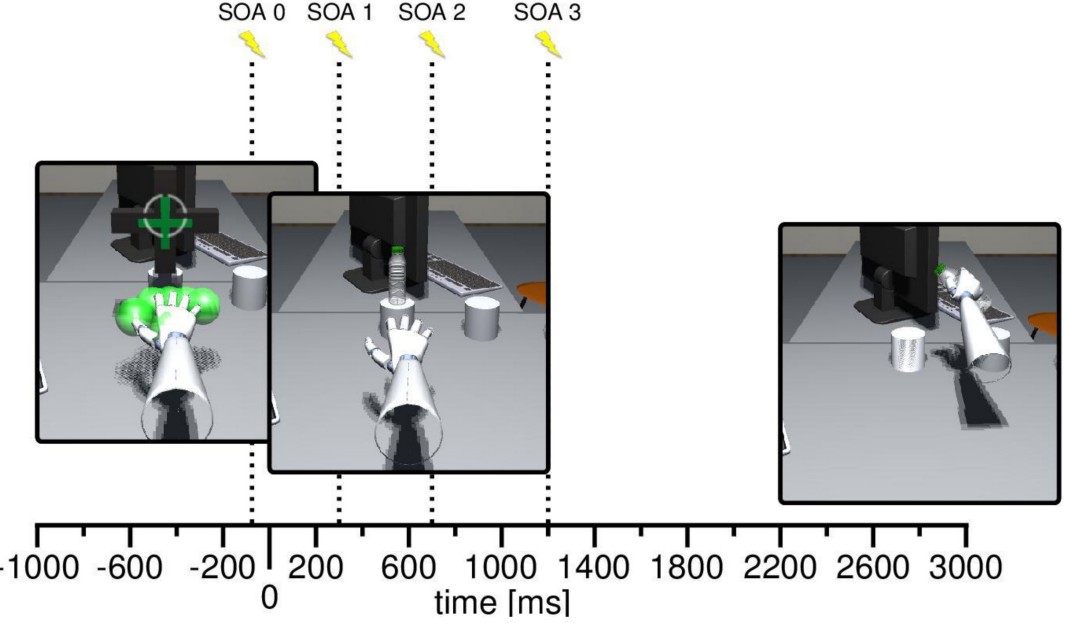

**Figure 2.** Mean time course of trials in Experiment 1 and 2. The stimulus onset asynchrony (SOA) between bottle and tactile and simultaneous visual stimulation varied from trial to trial. It could be applied 150 ms before the bottle appeared (SOA0, only in Experiment 1), 250 ms after the onset of the bottle (SOA1), at movement onset (SOA2), or after participants' hands traveled 50% of the distance to the bottle (SOA3). Stimulation onset is indicated by a yellow flash.

*2.4. Procedure*

At the beginning of the experiment, participants received a verbal instruction regarding the VR equipment. Then they were equipped with the vibration motors and familiarized with the stimulation.

Participants were then seated comfortably on an armchair and put on the HMD. Before the actual experiment, participants performed grasp training and trained the verbal response until they felt comfortable with both tasks. In the grasp training, participants performed the grasp-and-carry task without receiving a tactile stimulation. In the verbal response training, participants did not perform a grasping movement, but remained with their hand in the starting position.

In the actual experiment both tasks were combined in a dual-task paradigm. At the beginning of each trial, participants had to move their right hand into a designated starting position, consisting of red, transparent spheres indicating the required positions of the fingers and the palm. The spheres turned green when the respective fingers were in position (see Figure 2, bottom). Furthermore, participants had to maintain a stable gaze direction on a fixation cross by moving the crosshair in the center of their field of view above the fixation cross (the setup did not involve eye-tracking, but only head-tracking). Once both requirements were met for 1000 ms, the fixation cross as well as the visible markers of the initial hand position disappeared, and after 1000 ms a bottle appeared on the central pedestal. The bottle was either oriented upright or upside down. Participants were instructed to grasp the bottle with a power grasp, and to put it in an upright orientation within the target location. We did not explicitly instruct a supine (underhand) grasp, in case of inverted bottles. In line with the end-state comfort effect [37], however, all participants performed this kind of grasp.

Besides the grasp-and-carry task, participants had to discriminate which finger received a vibrotactile stimulation, and were thus asked to report the stimulated finger as fast as possible (by saying "index finger" or "thumb", i.e., in German, "*Zeigefinger*" or "*Daumen*") upon vibration detection. The onset of the tactile stimulation varied from trial to trial and the stimulation lasted for 250 ms. The visual distractor appeared at the same time at either the right or the left side of the bottle, lasting for 250 ms as well. Please note that the update frequency of the engine was 75 Hz. Timing precision of the visual distractor was tied to this update frequency, hence, the actual presentation time was 250 ms ± 13 ms. This also applied for the tactile stimulation, as it was controlled with the same update frequency. Depending on the bottle orientation, the combination of tactile stimulation and visual distractor was expected to yield different congruent and incongruent conditions with respect to the aCCE (see Figure 3). In the first experiment, we applied four different SOAs regarding the tactile stimulation. The stimulation could either occur 150 ms before the bottle appearance (SOA0), 250 ms after the bottle appearance (SOA1), at movement onset (SOA2), or during the movement; that is, after participants had travelled 50% of the distance to the target (SOA3). Visual distractors in case of SOA0 were shown at the same locations as in cases where the target was already visible. In the second experiment, only the latter SOAs (SOA1, SOA2, and SOA3) were applied. Participants had to respond verbally within 3000 ms of stimulation. Otherwise the trial was cancelled and considered an error trial.

The first experiment consisted of 192 trials, presented in a single block. The second experiment consisted of two blocks comprising 144 trials each. In one block the sensorimotor mapping remained unaffected, in the other one it changed from trial to trial. The variation of the sensorimotor mapping was realized by introducing a lateral shift of the visual hand model during the reaching movement towards the bottle. In half of the trials, a shift to the right was applied, while in the other half, the shift was to the left. In the case of a shift, participants had to compensate by moving their actual hand in the opposite direction. The random, trial-wise manipulation of the drift direction prevented the development of a general adaptation. Furthermore, the magnitude of the drift varied (three different magnitudes to each side: 5 cm, 7.5 cm, or 10 cm at the target location).

The experiments were self-paced and participants could pause between trials (but were requested to keep wearing the HMD). At the end of the main experiment, participants were asked to complete the igroup presence questionnaire (IPQ, [43]). The respective analyses can be found in the Supplementary Material (S3). The whole procedure took between 90 and 120 min, including preparation and training.

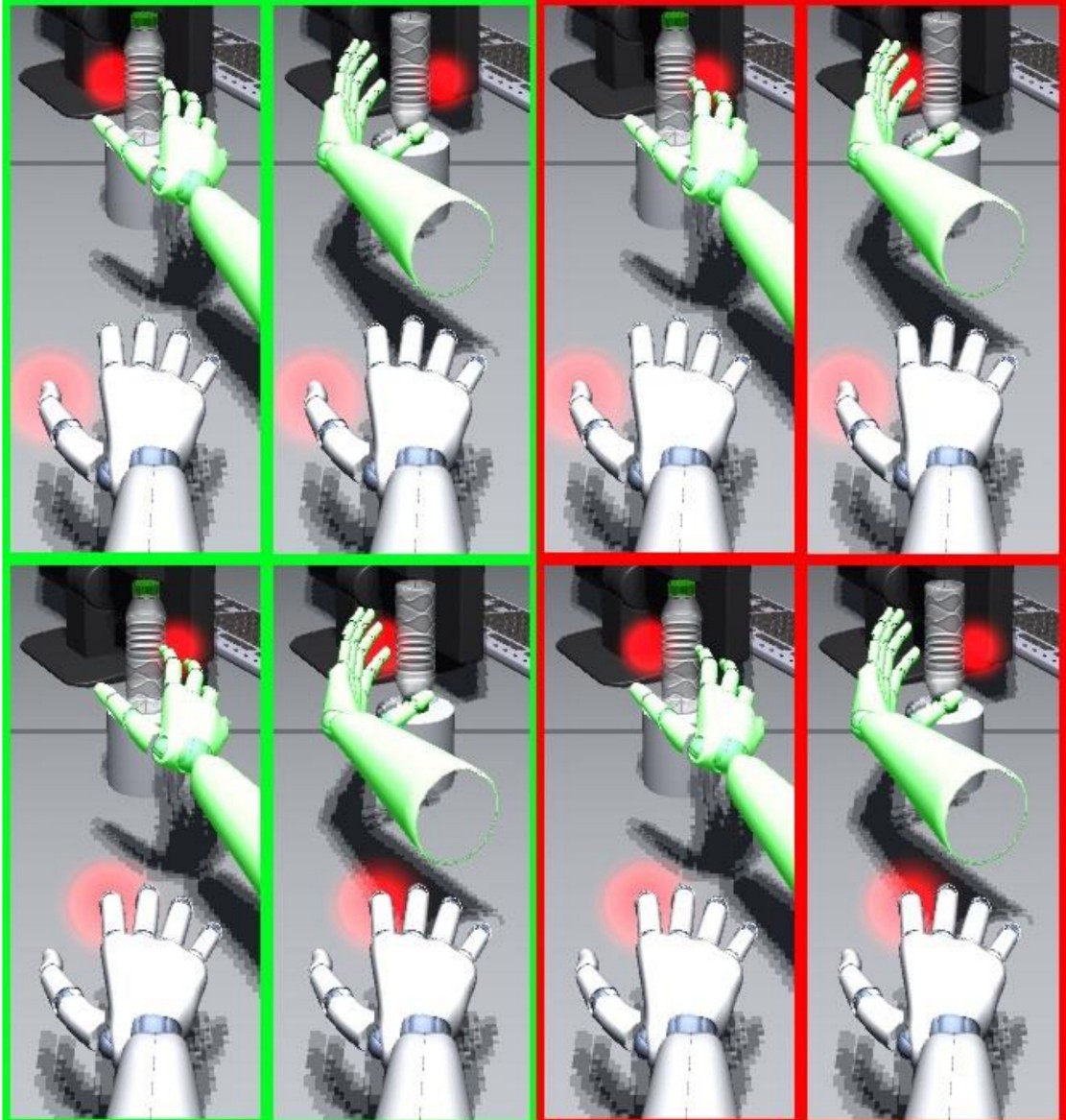

**Figure 3.** Stimulation conditions with respect to bottle orientation, visual distractor (red blob at the bottle), and vibrotactile stimulation (red blob at thumb or index finger). Green frames indicate stimulation conditions that are congruent with respect to the future finger position, red frames indicate incongruent conditions. Future hand positions are indicated by a green hand. Please note that the red blobs at the fingers were only added to visualize the vibrotactile stimulation and were not visualized in the experiment.

*2.5. Data Analysis*

In both experiments, we varied four factors across trials. First, the target bottle could be oriented upright or upside down (orientation). Second, the visual distractor could appear either on the left or the right side of the bottle (distractor). Third, the tactile stimulation could be applied either to the thumb or to the index finger (stimulation). Fourth, we varied the onset of the tactile stimulation and the visual distractor (SOA): 150 ms before the presentation of the bottle (Experiment 1 only; SOA0), 250 ms after presentation of the bottle (SOA1), at movement onset (SOA2), or after the hand traveled half-way to the bottle (SOA3). In Experiment 1, we repeated the 2 (orientation) × 2 (distractor) × 2 (stimulation) × 4 (SOA) factor combinations six times, yielding 192 trials. In Experiment 2, we introduced the variation of the sensorimotor mapping (mapping) across two experimental blocks. In the consistent block,

the sensorimotor mapping remained unaffected, while in the variable block, an incremental lateral offset to the visual hand was applied, depending on the distance to the bottle. To avoid the experiment lasting longer than necessary, SOA0 was not applied in Experiment 2. The resulting 2 (mapping) × 2 (orientation) × 2 (distractor) × 2 (stimulation) × 3 (SOA) factor combinations were repeated six times, yielding 288 trials (i.e., 144 trials per block). Trial orders were randomized. In Experiment 2, the block order was balanced across participants. The primary dependent measure was the verbal response time for naming the stimulated finger. The analysis presented here focuses on verbal RTs, additional analysis of error rates, movement onset times, and movement times can be found in the Supplementary Material (S2). Data from error trials (incorrect verbal response, or no response at all) were excluded from the analyses (4.3% in the first experiment and 1.9% in the second experiment). Seeing the small number of repetitions per cell, median response times were used to reduce outlier effects on the analysis. We did not apply any further outlier filtering procedures. Verbal RTs were analyzed using two repeated measures ANOVAs. The first one involved all factors, while the second one focused on the aCCE. Some conclusions regarding our hypotheses require the interpretation of null effects. This remains problematic when using typical frequentist analyses, hence, the respective tests were complemented with a Bayes factor analysis. We used a Cauchy prior for all Bayes factor analyses. We considered Bayes factors conclusive in favor for a hypothesis, if they were larger than 3, we considered them to be conclusive against a hypothesis if they were smaller than 0.3. Bayes factors between 1.5 and 3.0 were considered to provide anecdotic evidence for a certain hypothesis, while factors between 0.3 and 0.6 were considered as anecdotic evidence against a certain hypothesis. Bayes factors between 0.6 and 1.5 were considered inconclusive. All data and the scripts used for the analyses are available at the Open Science Framework (https://osf.io/ap7xt/).

*2.6. Congruency*

For our hypothesis, possible aCCEs were most relevant, which are reflected by three-way interactions between the factors orientation, distractor side, and stimulation (see Figure 3). For instance, in the case of an upright bottle, a tactile stimulation of the index finger along with a visual distractor on the right side of the bottle is congruent. To focus the analysis, we aggregated the response times in terms of incongruent and congruent conditions and analyzed the respective differences (incongruent-congruent) in Experiment 1 with a 2 (orientation) × 4 (SOA) ANOVA, and in Experiment 2 with a 2 (mapping) × 2 (orientation) × 3 (SOA) ANOVA. In these analyses. a significant, positive intercept indicates a significant aCCE (faster responses in congruent compared to incongruent conditions).

**3. Results**

*3.1. Experiment 1*

Verbal RTs from the 16 participants were analyzed with 2 (orientation) × 2 (distractor) × 2 (stimulation) × 4 (SOA) repeated measures ANOVAs. Verbal RT differences between incongruent and congruent conditions were further analyzed with a 2 (orientation) × 4 (SOA) repeated measures ANOVA. Only correct trials were included in the analysis. All reported post-hoc comparisons were submitted to a Holm-Bonferroni correction. The analyses were carried out with R [44] and the *ez* package [45]. In case of violations of the assumption of sphericity, *p*-values were submitted to a Greenhouse-Geisser adjustment. Bayes factor analyses were carried out using the BayesFactor package [46].

Analysis of verbal RTs yielded significant main effects for stimulation ($F(1,15) = 9.58$, $p = 0.007$, $\eta_p^2 = 0.39$) and SOA ($F(3,45) = 52.30$, $p < 0.001$, $\eta_p^2 = 0.78$), as well as significant interactions between orientation, distractor, and stimulation ($F(1,15) = 24.69$, $p < 0.001$, $\eta_p^2 = 0.62$), and the four-way interaction ($F(3,45) = 5.34$, $p = 0.003$, $\eta_p^2 = 0.26$; all remaining $p$'s $\geq 0.081$). Regarding the main effect of stimulation, participants were faster when responding to stimulations to the index finger ($Mdn_{index} = 729$ ms), as compared to stimulations to the thumb ($Mdn_{thumb} = 768$ ms). Seeing the higher

order interactions, a direct interpretation of this main effect is difficult. Further inspection of the main effect of SOA by means of post-hoc t-tests revealed that the response times decreased for later SOAs ($Mdn_{SOA0}$ = 871 ms; $Mdn_{SOA1}$ = 741 ms; $Mdn_{SOA2}$ = 706 ms; $Mdn_{SOA3}$ = 676 ms). This acceleration was significant for all pairwise comparisons (all adjusted p's < 0.018), except the difference between SOA2 and SOA3 (adjusted $p$ = 0.089). Again, due to the interactions, this main effect has to be treated with caution. To analyze the significant three and four-way interactions in further detail, we focused the analysis on the aCCE.

The 2 (orientation) × 4 (SOA) ANOVA of the verbal RT differences yielded a significant intercept ($F(1,15)$ = 24.69, $p$ < 0.001, $\eta p^2$ = 0.62), and a significant main effect of SOA ($F(3,45)$ = 5.34, $p$ = 0.003, $\eta_p^2$ = 0.26). There was no main effect for orientation ($p$ = 0.184) and no interaction between SOA and orientation ($p$ = 0.619). The intercept was positive ($M$ = 25 ms), implying an aCCE conforming to our hypothesis. Post-hoc analysis of the SOA effect showed that the aCCE was more pronounced for later SOAs. The aCCE at SOA3, that is, when the hand moved halfway towards the bottle, was significantly larger than in all other SOA conditions (all adjusted $p$s ≤ 0.043), whereas the aCCEs did not differ significantly from each other between the other SOA conditions (all adjusted $p$'s ≥ 0.99). To further probe the significance of the aCCE, all of the 2 (orientation) × 4 (SOA) mean differences were tested against a true mean of 0. For upright bottles, the means for SOA2 and SOA3 were significantly larger than 0 (adjusted $p$'s < 0.006). For bottles presented upside down, this was only the case for SOA3 (adjusted $p$ = 0.001). The results are shown in Figure 4, left panel.

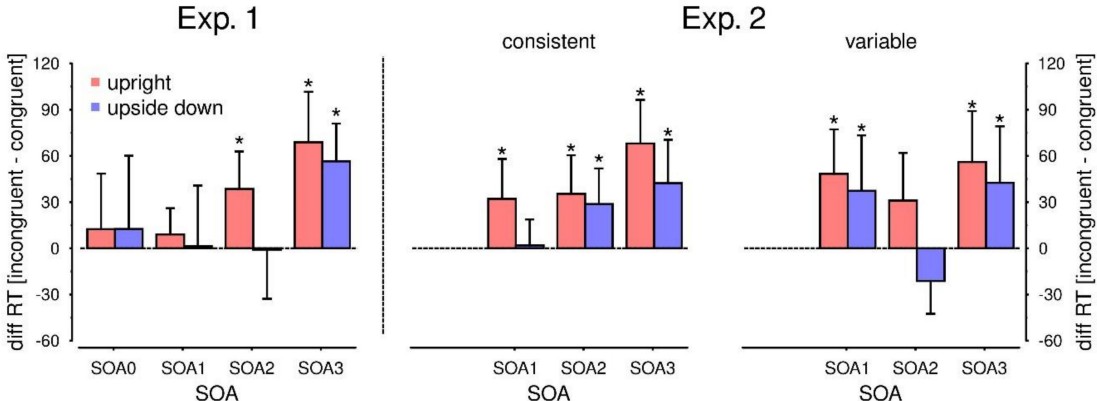

**Figure 4.** The aCCE per SOA as the difference in verbal RT between incongruent and congruent trials. Each difference was tested against a true mean of 0; significant differences are indicated by an asterisk. In the left panel, data from Experiment 1 is shown. Data from Experiment 2 is shown in the central (consistent mapping) and the right panel (variable mapping), respectively. In case of the variable mapping, the aCCE disappears at SOA2 (i.e., at movement onset). Error bars indicate the 95% confidence interval.

For our hypothesis that the aCCE would emerge during movement planning, the absence of an effect at SOA0 is crucial. To allow a more detailed assessment, we obtained the Bayes factors for all comparisons against 0. For SOA0, the estimated Bayes factor suggested that the data were 3.1 (upright bottle) times and 3.4 (upside down bottle) times more likely to be equal to 0 than larger than 0. For SOA1, the respective Bayes factors were 2.25 (upright bottles) and 3.9 (upside down bottles). For SOA2, the null hypothesis turned out to be highly unlikely in case of upright bottles ($BF_{01}$ = 0.1), but was favored for upside down bottles ($BF_{01}$ = 3.9). For SOA3, the Bayes factor suggested that the null hypothesis is highly unlikely for upright bottles ($BF_{01}$ = 0.01), as well as for upside down bottles ($BF_{01}$ = 0.01).

*3.2. Experiment 2*

The results from the first experiment showed a significant aCCE at SOA3. For the upright bottle, a robust aCCE was observed already at SOA2. According to our hypotheses, this reflects the unfolding, dynamic anticipatory behavior preparation and control process. This process is assumed to depend on the certainty of the predictions. To test this hypothesis, we varied the motion uncertainty while reaching for the bottle in Experiment 2. The additional factor mapping (variable and consistent) introduced variance in the visuomotor domain in one block of the experiment. We analyzed verbal RTs from 21 participants with a repeated measures ANOVA according to the 2 (mapping) × 2 (orientation) × 2 (distractor) × 2 (stimulation) × 3 (SOA) factorial design.

Analysis of verbal RTs yielded significant main effects for orientation ($F(1,20) = 8.01$, $p = 0.010$, $\eta_p^2 = 0.29$) and SOA ($F(2,40) = 26.06$, $p < 0.001$, $\eta_p^2 = 0.57$). Participants responded faster in case of bottles presented upright ($Mdn = 726$ ms vs. 741 ms). As in the first experiment, verbal RTs decreased for later SOAs ($Mdn_{SOA1} = 774$ ms; $Mdn_{SOA2} = 733$ ms; $Mdn_{SOA3} = 693$ ms). This acceleration was significant between all SOA conditions (all respective $p$'s < 0.001). This general pattern was modified by various interactions. Regarding the two-way interactions, the interaction between orientation and stimulation was significant ($F(1,20) = 5.40$, $p = 0.031$, $\eta_p^2 = 0.21$), as well as the interaction between distractor and stimulation ($F(1,20) = 10.20$, $p = 0.005$, $\eta_p^2 = 0.34$), and the interaction between orientation and SOA ($F(2,40) = 4.38$, $p = 0.019$, $\eta_p^2 = 0.18$). The interaction between orientation and stimulation was driven by significantly faster responses in cases of upright bottles and stimulations on the index finger ($Mdn = 725$ ms), compared to upside down bottles and index finger stimulations ($Mdn = 748$ ms, respective $p = 0.04$). Regarding the interaction between SOA and orientation, significant differences between orientations were only observed at SOA1 ($Mdn_{upright} = 759$ ms, $Mdn_{upside\ down} = 789$ ms, $p = 0.017$), while the differences decreased at SOA2 ($Mdn_{upright} = 726$ ms, $Mdn_{upside\ down} = 740$ ms, $p = 0.224$) and SOA3 ($Mdn_{upright} = 694$ ms, $Mdn_{upside\ down} = 694$ ms, p = 0.997). None of the differences regarding the interaction between distractor and stimulation remained significant after adjusting for multiple comparisons. Most importantly, the three-way interaction between orientation, distractor, and stimulation was significant ($F(1,20) = 41.76$, $p < 0.001$, $\eta_p^2 = 0.68$), as well as the interaction between SOA, orientation, distractor, and stimulation ($F(2,40) = 5.55$, $p = 0.007$, $\eta_p^2 = 0.22$), and the interaction between SOA, orientation, distractor, stimulation, and mapping ($F(2,40) = 3.94$, $p = 0.027$, $\eta_p^2 = 0.16$). There were no further significant effects (all remaining $p$'s ≥ 0.123). For the results to support our hypothesis that the activated spatial remapping indeed interfere with the aCCE at movement onset, it is critical that the verbal RTs do not differ a priori at movement onset for the mapping conditions. We tested this assumption separately for both bottle orientations using Bayes factors. For the upright bottles, the Bayes factor suggested that there was no difference between the verbal RTs in the two mapping conditions at movement onset ($BF_{01} = 2.6$). For upside down bottles, the Bayes factor slightly favored the null hypothesis that there is no difference between the mapping conditions ($BF_{01} = 1.6$). To further analyze the significant higher-order interactions, we focused the analysis on the aCCE.

The 2 (mapping) × 2 (orientation) × 3 (SOA) ANOVA of the verbal RT differences yielded a significant intercept ($F(1,20) = 41.76$, $p < 0.001$, $\eta_p^2 = 0.68$), main effects for orientation ($F(1,20) = 10.20$, $p = 0.005$, $\eta_p^2 = 0.34$) and SOA ($F(2,40) = 5.55$, $p = 0.007$, $\eta_p^2 = 0.22$), as well as a significant two-way interaction between SOA and mapping ($F(2,40) = 3.94$, $p = 0.027$, $\eta_p^2 = 0.16$). No further main effects or interactions reached significance (remaining $p$'s ≥ 0.123). As expected, the intercept was positive ($M = 34$ ms). Post-hoc analysis of the SOA effect showed that the aCCE was most pronounced for the latest SOA. The aCCE at SOA3 was significantly larger than at SOA2 ($Mdn_{SOA2} = 18$ ms, $Mdn_{SOA3} = 52$ ms; $p = 0.018$). No further comparisons yielded significant results (adjusted $p$'s ≥ 0.099). According to our hypothesis, we expected differences between the mapping conditions, especially at movement onset. Post-hoc comparisons of the three SOA conditions for the two mappings showed a significant difference at SOA2, that is, at movement onset ($t(20) = -2.64$, $p = 0.047$). There was no significant difference between the mapping conditions at SOA1 ($t(20) = 1.94$, $p = 0.133$) or SOA3 ($t(20) = -0.35$, $p = 0.726$). A complementary analysis using Bayes factor supported this interpretation.

For SOA2, the Bayes factor supported the alternative hypothesis that the medians differ between the mapping conditions ($BF_{10}$ = 3.49); this was not the case for SOA3 ($BF_{10}$ = 0.24). For SOA1, the Bayes factor was indecisive ($BF_{10}$ = 1.09). Again, to further probe the significance of the aCCE, all of the 2 (mapping) × 2 (orientation) × 3 (SOA) mean differences were tested against a true mean of 0. The results are shown in Figure 4 in the central and right panels.

In summary, while the results confirm the observations made in Experiment 1, they additionally revealed a reduced aCCE in the case of the variable mapping at movement onset. Here, the respective tests against 0 were not significant, however, in the case of upright bottles, significance was missed only narrowly ($t$(20) = 2.07, $p$ = 0.051). Again, to allow a valid assessment of the null hypotheses that the different means do not differ from 0, a Bayes factor analysis was performed. For the consistent mapping, the Bayes factor supported the alternative hypothesis for upright bottles at SOA1 ($BF_{10}$ = 3.16), SOA2 ($BF_{10}$ = 6.07), and SOA3 ($BF_{10}$ = 397.65). For upside down bottles, the Bayes factor favored the alternative hypothesis for SOA2 ($BF_{10}$ = 3.16) and SOA3 ($BF_{10}$ = 8.71), but not SOA1 ($BF_{10}$ = 0.23). For the variable mapping, the Bayes factor supported the alternative hypothesis for upright bottles at SOA1 ($BF_{10}$ = 17.63) and SOA3 ($BF_{10}$ = 19.17), while it remained indecisive at SOA2 ($BF_{10}$ = 1.35). For upside down bottles, the Bayes factor supported the alternative hypothesis at SOA3 ($BF_{10}$ = 2.37), but remained almost indecisive at SOA1 ($BF_{10}$ = 1.53) and SOA2 ($BF_{10}$ = 1.35).

## 4. Discussion

In two experiments, we investigated anticipatory cross-modal congruency effects (aCCEs) between visual stimuli at a future finger position and tactile stimulation of the finger at its current position. Participants had to grasp bottles in VR. Meanwhile, vibrotactile stimulations were applied before or during movement execution. In general, aCCEs were more pronounced for stimulations at later SOAs, in line with previous results [33–35,38]. Thus, the results from the first experiment show that peripersonal hand space (PPHS) is remapped onto the future hand location and posture in advance of a virtual goal-directed grasping movement. Our main aim was to shed light on the actual remapping mechanisms. According to theories of event-predictive active inference, the remapping is realized by a generative process that predicts the likelihood of future sensations. This estimation should depend on expected uncertainties in the sensorimotor mapping. Thus, the second experiment aimed at investigating the consequences of planning uncertainties on the aCCE. To do so, we manipulated uncertainty by introducing a discrepancy between seen and actual hand position during prehension. The direction and amount of this discrepancy only became apparent at movement onset. As a result, the aCCE was reduced at movement onset, but persisted for earlier and later stimulations. Hence, the dynamics of the aCCE were influenced by expected control uncertainties.

Overall, the results support theories of probabilistic, event-oriented, active inference. PPHS was adaptively remapped onto the next event boundary, yielding multisensory interactions at the future hand location in anticipation of the next control-critical sensations [15]. In the experiments, the event boundary of establishing a successful grasp was the primary goal, which leads to the anticipatory remapping of PPHS. However, the approaching hand event needs to be controlled successfully before the boundary is reached. Thus, when control uncertainty was increased in the variable mapping block in Experiment 2, the sensory processing focus apparently switched temporarily to the current hand position and orientation, diminishing the aCCE accordingly. This distinction also dovetails with other approaches that consider movement control as a feedback system that aims to minimize localization errors [40,47]. VR setups combined with real-time motion tracking seem well-suited to elaborate this interpretation even further. The proposed focus on the next event boundary could be, for example, verified further by instructing sequential tasks. According to our theory, the active inference process should always focus on the next upcoming event boundary but it should also have pre-activations of successive event boundaries in mind, and meanwhile, ensure that the currently controlled event successfully unfolds.

While the presented results are generally in line with the outlined theoretical framework, two aspects of the data pattern, namely the aCCE for early SOAs and the effect of bottle orientation on the aCCE, remain problematic. Furthermore, even if the active inference framework provides a formal account for the observed results, the neurophysiological basis remains open. In our first experiment, we did not observe an aCCE at SOA1, while the aCCE was present at SOA1 in the second experiment (see Figure 4). We observed a similar difference between experiments with different SOAs in our previous study applying pantomimic movements and argued that this difference might be due to strategic differences in the dual-task scheduling [35]. In our first experiment, stimulation occurred before movement onset in 50% of the trials (SOA0 and SOA1), while this was only the case in 33% of the trials in the second experiment (SOA1). This might have encouraged earlier movement planning—and accordingly earlier remapping of PPHS—in the second compared to the first experiment. However, even if such differences in dual-task scheduling account for the differences regarding the SOAs, they cannot account for the differences in the aCCE between upright and upside down bottles. For upside down bottles, the aCCE was weaker in both experiments, and occurred later, at least in the first experiment and in the consistent condition in the second experiment (see Figure 4). Regarding the verbal RTs, responses to upright compared to upside-down bottles were faster in both experiments (19 ms in Experiment 1, 15 ms in Experiment 2, this difference was only significant in Experiment 2). This implies a general advantage for bottles presented upright. The upright orientation might be considered as the canonical orientation of a bottle, and it has been shown that objects presented in a canonical manner are recognized faster [48]. Hence, it is tempting to assume that the higher familiarity with upright bottles facilitates remapping, and accordingly yields a stronger aCCE. However, this should also foster movement planning and result in earlier movement onsets. This is not the case (see Supplementary Material), hence it remains questionable if the observed difference is due to familiarity with the bottle orientation. Other studies on multisensory interaction during goal-directed actions [33,34,38] used oriented bars as grasping targets. A direct comparison between those and the virtual bottles would be necessary to investigate stimulus-specific effects on aCCEs. As an alternative explanation, it might be the case that the initial hand posture affected the aCCE. In our experiments, the initial hand posture matched with an overhand grasp more than an underhand grasp. Again, a direct comparison of different initial hand postures would be necessary to investigate if the aCCE is modulated by the postural discrepancy between initial and anticipated hand posture.

The finding of an anticipatory remapping of PPHS onto the next event boundary in the service of goal-directed action fits well with the perspective of event-predictive cognition and according formalizations of active inference [10,12]. This perspective is also closely related to the stochastic bubble proposed by Noel et al. [40], as it considers PPHS as a probabilistic representation used to predict the likelihood of multisensory input. However, the neurological foundations of this anticipatory remapping remains open.

With respect to vision, there is evidence for anticipatory remapping of receptive fields in case of saccadic eye-movements [49]. Apparently, the receptive field of retinocentric neurons in the lateral parietal cortex switches to the target location of a saccade just before the actual eye-movement is carried out. This seems like the realization of a spatial prediction that tunes sensory processing to expected input. Cléry and colleagues [50] discussed a similar mechanism in PPS. In their study, participants were requested to detect tactile stimulations on their cheeks. If the tactile stimulation was preceded by a visual stimulus looming in front of their face, the detection rate was increased, especially when the apparent impact location of the visual stimulus matched the location of the tactile stimulation. It seems that vision and touch were not only integrated, but that the visual stimulus facilitated an impact prediction, which enhanced processing for tactile stimuli at a corresponding spatial location. According to Cléry et al. [50], impact prediction is likely to be realized within the ventral intraparietal area by neurons that integrate self-motion and visual-motion cues.

While these findings imply that PPS is involved in predictive processing for defensive purposes, similar mechanisms might be engaged in the preparation and control of goal-directed grasping

movements. The multisensory nature of PPS (including motor codes [51]) offers itself as a means to predictively map different, temporarily associated, modal sources of information (such as visual and tactile information) onto each other. Moreover, when projected in time, PPS can be used to anticipate upcoming sensory interactions. Combined with event-predictive, active inference, it appears that such predictive encodings indeed pre-activate anticipated stimulus interactions (fingers touch the bottle). However, the inference processes also depend on expected uncertainties, dynamically focusing on those mappings and associations that are critical for ensuring the execution of successful, goal-directed interactions with the environment. In conclusion, it appears that which aspects of the future our mind is anticipating at a certain point in time depends on our current goals, the available predictive structures, as well as on predicted uncertainties about the controllability and application success of these structures in the light of the current circumstances.

**Supplementary Materials:** The following are available online at http://www.mdpi.com/2411-5150/3/2/15/s1. The R files and the data for the Monte Carlo simulation of the power analysis can be found in Supplementary 1. Supplementary 2 contains additional analyses of movement onset times, movement times, and error rates. Supplementary 3 contains the analyses of the igroup presence questionnaire (IPQ).

**Author Contributions:** M.V.B., A.B., and J.L. developed the study concept. J.L. implemented the hardware and software setup and performed the testing and data collection. J.L. performed the data analysis under the supervision of M.V.B., J.L. and M.V.B. drafted the manuscript. A.B. provided critical revisions. All authors approved the final version of the manuscript for submission.

**Funding:** Funding from the Feodor-Lynen Grant of the Humbolt Foundation is gratefully acknowledged.

**Conflicts of Interest:** The authors declare no conflict of interest.

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
