# Peer review of "Hands Ahead in Mind and Motion: Active Inference in Peripersonal Hand Space"

_2411-5150_

Round 1
Reviewer 1 Report
The current manuscript by Lohmann et al describes an interesting experimental manipulation within the bottle grasping setup of last year’s paper by Belardinelli et al. First, they show a replication of the aCCE in VR, or at least of the results in the second experiment of Belardinelli et al., who showed the bottles on a touch screen. Second, by manipulating the movement of the avatar arm in VR, giving it a directional offset during movement, they disturb this effect at movement onset.
I am a bit confused by this manuscript. On the one hand, I think it investigates a sound, testable prediction based on the aCCE/bottle paradigm. If this aCCE is indeed based on predictive constructs, changing predictability on either side of the equation (bottle or hand) should influence aCCE. While not a huge scientific breakthrough, I think these kind of theoretical replications are important and should be given a platform.
The setup seems adequate to investigate the question (if a control condition is added) and I like the unpredictable direction of a movement mismatch in the visual feedback as an experimental manipulation. Simple, yet fit. On the other hand, the manuscript itself is rather meagre and the results a bit unclear. I think the theoretical framing, analyses and interpretation are insufficient. It will have to be at least very substantially rewritten before I would find it suitable for publication.
That is, if the authors can reassure me that the seen effects are not simply due to the double task being harder in some conditions than in others. For now, I would recommend a major revision.
Major concern: Possible confounding factor
Participants had to move their hand to a virtual target, while responding to a tactile task. In the variable condition, I would think reaching to the virtual target is harder, as participants will have to dynamically adjust their movement to the visual feedback. When stimuli are presented at movement onset, and last for 250 ms. during the movement, I would expect in this condition less attention is available for the visual distractors, and the aCCE decreases. This is exactly the timepoint where the double task is the hardest: while figuring out the ofset. (Double task interference is supported by the aCCE mainly decreasing in the inverted bottle condition, which would require a more complex hand movement.) Also, the authors claim the aCCE in exp 1 increases with larger SOAs, as would be expected as this is when the hand movement needs to be dynamically altered online.
I think this experiment really needs a control condition to rule out a simple variance in the attentional load of the double task being the driving force behind the results.
Additionally, the manuscript really needs to discuss spatial attention with respect to the current setup when using a spatial distractor task like this.
Which brings me to:
Major concern: Theoretical framing
The theoretical framing of the current experiment seems to me chaotic and not thorough enough. Several important topics and references are missed (predictive coding of PPS, approaching stimuli in PPS, spatial attention in these cases, PPS as a prior in multimodal coupling). Noel et al’s paper last year in Nature isn’t cited. Peripersonal space is sometimes described as an extension of the body representation or even constituted of “spatial body representations” (page 2, line 66)(please cite the relevant papers too), further on as an interface for body-object interaction, and also as a network of (spatial? External?) body-part centered representations while the authors actually appear to see it as a visuo-proprioceptive Bayesian inference problem like Noel et al in 2018. While all these views have been fruitful approaches in PPS research and may supplement each other, hopping from one view to another within one paper makes for a rather chaotic impression and in part causes the hypothesis to be unclear. (note: not the prediction, which is clear). (Similarly by the way, I think hopping to and from a SOA-based view of the data to “earlier and later time points” is confusing)
While I can see how you would reach the prediction that increasing uncertainty over the hand movement during planning would influence aCCEs, no mechanisms are discussed enough. Several are mentioned though. On page 2, the topic goes to “remapping”, which could perhaps be the underlying mechanism the authors hypothesise, but is not really explained. In the discussion and introduction, it remains unclear what exactly is expected to be remapped into what and what happens to the different reference frames involved. (Also, I would say claiming something is “still being debated (page 2, line 69) would require a more recent reference than one from 2012.)
Page 10, line 361 mentions the motivation for exp 2 is to investigate whether aCCE in the normal bottle-setup is caused by event-predictive inference processes rather than mere shifts in spatial attention. However, this motivation is not discussed in the introduction. Moreover, exp 1 as well as the original paper already suggests specific event-related spatial matches are causing this effect. While exp 2 investigates a clear, testable prediction, it is not discussed why this is more than a theoretical replication, if it is.
Major concern: Analyses
In all, it seems the analyses and description of them need more careful consideration. The authors for instance now claim (page 9 line 293) “For upright bottles, all means except for SOA0 and SOA1 were significantly larger than 0...” which is technically true, but a misleading way of framing when there are 4 conditions. 2 out of 4 is not “all but these”, that’s half. This is an example, the results need to be checked on the text doing the data justice.
Moreover, conclusions are based on *not* finding an effect in the variable condition at SOA2. This is questionable using frequentist statistics, made even more questionable by the fact that significance was missed only very narrowly in the case of upright bottles. I would not consider this interpretation valid unless they redo the analysis using Bayesian statistics, and a decent discussion of the fact that it is mainly the condition with the upside down bottles that loses the aCCE. As well as a fair description of the results.
Major concern: Interpretation of the results
The current results are barely discussed. Many patterns present in the data are not further interpreted in the discussion. While the setup allows for SOAs to be investigated with respect to the timing of the hand movement (movement planning vs execution) and the double task involves an online adjustment of the movement, the data is hardly discussed as such. On page 9 line 304, results are said to reflect dynamic anticipatory behavior preparation while being present at the start of and during the execution of the movement rather than the planning. See also the abstract. Also, see major concern 1 for why this may not match the prediction.
Instead, the discussion starts by focusing on remapping. The manipulation of visuo-proprioceptive mapping is argued to cause the aCCE to vanish at movement onset (see concerns about data analysis), but persist at earlier and later stimulations. However, this is a bit unfair framing, as in experiment 1 no aCCE is seen at all at earlier time points. This difference between exp 1 and 2 is not discussed.
Finally, the conclusion on page 11, line 396-399 is not based on the current results.
Other analysis points:
# As the raw RTs become shorter with longer SOAs (temporal preparation effect possibly? I don’t think it is discussed?) perhaps aCCE should be analysed as a percentage change in RT. Please explain why this is not done.
# The authors could add Bayesian informative hypotheses comparison to substantiate the claim on page 10 line 355 that the aCCE increases with later stimuli in this experiment. I’d still be hesitant to call it a temporal buildup, instead of referring to the anchor points of the SOAs with respect to the hand movement.
# While medians are more robust to outliers than means, 4 datapoints per condition is not a lot. There could still be instances where >2 trials per condition recorded no reliable RT. I generally agree with keeping as much data as possible, as outlier detection can become subjective. But what happened to trials with responses before the stimulus? Or within 50-100 ms? Or after for instance 4000 ms?
# Main effects are hard to interpret when there are 3- or even 4-way interactions present, please refrain from discussing them as if the interaction were not there.
# RTs are analysed but not depicted in a graph or interpreted, perhaps rather only to be checked for the expected interaction. If this is the sole reason for these analyses, they do take up rather a disproportionate amount of text.
Other methodological points:
# How does this variability manipulation influence ownership and feeling of agency over the VR hand, thereby influencing in a way tool us? There is a lot of literature on tool use (and the fact that you have to really use them in these setups) on visuotactile congruency effects (see for instance the work by Nicholas Holmes).
# What is the temporal delay in the VR image? Does that differ between conditions?
# Could the second pedestal act as a rival potential target, or as an obstacle while grasping? (see obstacle avoidance literature).
# timing precision of the visual and tactile stimuli had a variation of 13 ms due to update frequency. Were both stimuli (visual and tactile) however presented at the exact same moment? Vibration motors generally have a startup time before reaching full strength, making their timing somewhat difficult.
# The way in which fixation and a correct starting position are reached is well-designed. I think a lot of time and effort has been put in programming this setup.
Author Response
We wish to thank the reviewer for the detailed assessment of our manuscript. We hope that we adressed all major issues, please find our point-by-point responses in the uploaded document.

Reviewer 2 Report
Lohmann and colleagues present two experiments. In the first they show that peri-personal space (PPS) as indexed by remaps before making contact with an object. In a second experiment they largely replicate the first, while also demonstrating that this effect is sensitive to perturbation in hand position.
Overall I find the study well executed, I do however have a concern about the authors’ interpretation of the main result and believe the authors can do a more accurate job at describing previous research in this field.
My major comment is about interpreting changes in hand location as changes in uncertainty. On the time-course of a single trial, this seems like a change of mean to me, not a change of variance. There is no uncertainty about where the virtual hand is; it is where it is presented. On the other hand, I do think this manipulation introduces two other variables. First, it changes the relative position between the visually presented virtual hand, and the real hand. This change can certainty change how much the virtual hand is embodied, and hence maybe even the size and shape of the real hand PPS (Noel et al., 2015, Cognition; Salomon et al., 2017, Cognition). Further, it also changes the distance between the hand and the bottle. PPS is mostly thought of as spatial feature, thus how do the authors know that they just haven’t simply changed the distance, which will definitely impact PPS?
I believe there are quite a few references the authors either omit entirely, or don’t use the best reference possible, which misrepresents PPS research. First, there are a number of studies showing PPS remapping before movement onset, and this should be made very very clear (e.g., Patane et al., Journal of Cognitive Neuroscience). Further, there are a host of articles suggesting that PPS is involved in prediction (Clery et al., 2015, Journal of Neuroscience; Noel et al., 2018, Annals of the New York Academy of Science).
The authors claim that PPS is multisensory and cite Holmes et al., and Brozzoli et al. It is true that PPS is multisensory, yet the best examples of this comes from Avillac et al., 2007, Journal of Neuroscience and Bernasconi et al., 2018, Cerebral Cortex.
I find the description of the timing between events a bit confusing, since some of these times are described from the time of bottle presentation onset, and some are presented as from time of movement. Please choose one and express all other timing relative to the same endpoint.
Author Response
We wish to thank the reviewer for the detailed assessment of our manuscript and the pointers to the additional literature. We hope we could adress all raised issues, please find our point-by-point responses in the attached document.

Round 2
Reviewer 2 Report
The authors have addressed all of my concerns.